# May EPH/Ephrin Targeting Revolutionize Lung Cancer Treatment?

**DOI:** 10.3390/ijms24010093

**Published:** 2022-12-21

**Authors:** Iason Psilopatis, Ioannis Karniadakis, Konstantinos Stylianos Danos, Kleio Vrettou, Kleita Michaelidou, Konstantinos Mavridis, Sofia Agelaki, Stamatios Theocharis

**Affiliations:** 1First Department of Pathology, Medical School, National and Kapodistrian University of Athens, 75 Mikras Asias Street, Bld 10, Goudi, 11527 Athens, Greece; 2Department of Gynecology, Charité—Universitätsmedizin Berlin, Corporate Member of Freie Universität Berlin and Humboldt—Universität zu Berlin, Augustenburger Platz 1, 13353 Berlin, Germany; 3Second Department of Propaedeutic Surgery, “Laiko” General Hospital, 17 Agiou Thoma Street, 11527 Athens, Greece; 4Laboratory of Translational Oncology, School of Medicine, University of Crete, Vassilika Vouton, 71003 Herakleion, Greece; 5Institute of Molecular Biology and Biotechnology, Foundation of Research and Technology-Hellas, 70013 Herakleion, Greece; 6Department of Medical Oncology, University General Hospital of Herakleion, Vassilika Vouton, 71110 Herakleion, Greece

**Keywords:** ephrin receptor, EPH, ephrin, lung cancer, treatment, precision medicine

## Abstract

Lung cancer (LC) is the leading cause of cancer death in the United States. Erythropoietin-producing hepatocellular receptors (EPHs) comprise the largest receptor tyrosine kinases (RTKs) family in mammals. EPHs along with their ligands, EPH-family receptor-interacting proteins (ephrins), have been found to be either up- or downregulated in LC cells, hence exhibiting a defining role in LC carcinogenesis and tumor progression. In their capacity as membrane-bound molecules, EPHs/ephrins may represent feasible targets in the context of precision cancer treatment. In order to investigate available therapeutics targeting the EPH/ephrin system in LC, a literature review was conducted, using the MEDLINE, LIVIVO, and Google Scholar databases. EPHA2 is the most well-studied EPH/ephrin target in LC treatment. The targeting of EPHA2, EPHA3, EPHA5, EPHA7, EPHB4, EPHB6, ephrin-A1, ephrin-A2, ephrin-B2, and ephrin-B3 in LC cells or xenograft models not only directly correlates with a profound LC suppression but also enriches the effects of well-established therapeutic regimens. However, the sole clinical trial incorporating a NSCLC patient could not describe objective anti-cancer effects after anti-EPHA2 antibody administration. Collectively, EPHs/ephrins seem to represent promising treatment targets in LC. However, large clinical trials still need to be performed, with a view to examining the effects of EPH/ephrin targeting in the clinical setting.

## 1. Introduction

Lung cancer (LC) represents the second most common malignancy and the leading cancer death cause in the United States, with the mean age of patients upon diagnosis at 70 years of age [1]. According to the American Cancer Society, about 236,740 new cases of LC will be diagnosed and about 130,180 patients will die from LC in the United States in 2022 [1]. LC is divided into two distinct histologic classes: the more aggressive small-cell LC (SCLC) and the more common non-small-cell LC (NSCLC) [2]. The main NSCLC subtypes are adenocarcinoma, squamous cell carcinoma, and large cell carcinoma [2]. Pulmonary symptoms of LC predominantly include cough, hemoptysis, dyspnea, or chest pain, while extrapulmonary symptoms are constitutional symptoms, compression of neighboring structures, and paraneoplastic syndromes [3]. The diagnostic evaluation of LC includes, in addition to a physical examination, a chest X-ray as a first-line imaging study, and a computed tomography (CT) scan of the chest in cases of abnormal X-ray. Definite diagnosis always requires biopsy of the tumor mass. Advanced studies for LC staging involve a CT of the thorax and abdomen, brain imaging, and positron emission tomography (PET)-CT [4]. For patients with non-metastatic resectable NSCLC, surgical excision, followed by chemotherapy, represents the first-line therapy. Non-surgical patients are mostly treated with chemotherapy plus radiotherapy, and possibly immunotherapy, whereas molecular-targeted treatment may be offered in cases of targetable mutation [5]. Limited-stage SCLC always requires systemic therapy and concurrent radiotherapy, whereas chemotherapy with or without immunotherapy is the proposed therapeutic regime for extensive-stage SCLC [6].

Erythropoietin-producing hepatocellular receptors (EPHs) comprise the leading subfamily of receptor tyrosine kinases (RTKs), which bind the membrane-bound proteins, ephrins [7]. EPHs are categorized into two subgroups, EPHAs and EPHBs, depending on their structural homology and preferential binding affinities to their respective ligands, ephrin-A and ephrin-B [8,9]. Specifically, EPHA binds ephrin-A via a glycosylphosphatidylinositol anchor on a plasma membrane, whereas ephrin-B interacts with EPHB via a transmembrane domain [10]. In humans, nine EPHA receptors (EPHA1-8, 10), which interact with five ephrin-A ligands (ephrin-A1-5), along with five EPHB receptors (EPHB1-4, 6), which bind three ephrin-B ligands (ephrin-B1-3), have been defined [11]. The EPH–ephrin interaction prompts both forward signaling in the receptor-expressing cell, as well as backward signaling in the ephrin-bearing cell [12]. EPHs/ephrins show a wide expression in various cell types, and influence diverse physiological functions related to cell migration, cell–cell or cell–matrix interactions, or (lymph-) angiogenesis, thus potentially exerting both tumor-promoting and tumor-suppressive properties [12,13,14,15,16,17,18,19,20].

Over the past decade, a large number of original articles has been published on the role of different members of the EPH/ephrin system in the pathogenesis and progression of LC; moreover, these studies described their implication in (N)SCLC molecular pathways, immune evasion, metastasis, recurrence, and clinicopathological features, as well as patient survival and prognosis [21,22,23,24,25,26,27,28,29,30,31,32,33,34]. Recently, both Kou et al. and Anderton et al. extensively reviewed the implication of diverse EPHs/ephrins in LC and outlined the tumor-promoting effects of EPHA1, EPHA2, EPHA4, EPHA5, EPHA7, EPHB3, EPHB4, ephrin-A3, and ephrin-B2, as well as the tumor-suppressive effects of EPHA3, EPHB6, and ephrin-B3 [35,36]. Inspired by these observations, and given the current advancements in the field of targeted therapy, several study groups have recently attempted to investigate the feasibility of targeting the EPH/ephrin system in LC. The present review summarizes the results of all available relevant studies and presents newly developed agents that target the various members of the EPH/ephrin system in LC.

## 2. Available EPH/Ephrin-Targeting Agents in LC

Since their clinical introduction in 2001, tyrosine kinase inhibitors (TKIs) represent potent pharmacologic agents that target the active site of RTKs and, thereby, either reversibly or irreversibly, inhibit the phosphorylation of intracellular targets [37]. Imatinib became the first Food and Drug Administration (FDA)-approved TKI for the treatment of Philadelphia chromosome–positive chronic myeloid leukemia-targeting Bcr-Abl, thus not only revolutionizing treatment for these patients but also laying the foundation for the use of targeted therapeutics [38]. To date, the FDA has approved a large number of TKIs for mostly epidermal growth factor receptor (EGFR)-mutated NSCLC [39]. Given that EPHs represent a subgroup of RTKs, consequently, dasatinib, osimertinib, ensartinib, ALW–II–41-27, and XL647, as well as PKC 412, seem to represent promising TKIs for EPH-expressing LC.

Short interfering RNAs (siRNAs) comprise a class of regulatory small RNA molecules that modify the stability or translational efficacy of messenger RNAs (mRNAs), thus silencing any disease-related genes in a sequence-specific manner [40]. siRNAs are currently widely used in diverse types of cancers, such as pancreatic, breast, colorectal, ovarian, hepatocellular, gastric, and cervical cancer [41]. In LC, siRNAs have already shown promising effects in preclinical and early clinical evaluations of both SCLC and NSCLC [42], thus paving the way for their employment in EPH-expressing LC.

By precisely binding to antigens on the surface of cancer cells, monoclonal antibodies (mAbs) represent laboratory-produced proteins that provoke long-lasting anti-cancer immune responses [43]. In the same context, immunoliposomes can be generated by antibody coupling to the liposomal surface, thus facilitating active tissue targeting [44]. The FDA has, to date, approved several mAbs for the treatment of both SCLC and NSCLC [45]. EPHs/ephrins, in their capacity as membrane-bound proteins, may, consequently, also serve as feasible targets of mAbs or immunoliposomes in LC.

Albumin represents a highly soluble and stable drug delivery system with a long circulatory half-life and high intratumoral accumulation due to its enhanced permeability and retention effect [46]. After its approval in 2005, Abraxane became the first and sole FDA-approved paclitaxel–albumin nanoparticle for the treatment of metastatic breast cancer, locally advanced or metastatic NSCLC, and metastatic pancreatic adenocarcinoma [47]. In EPH/ephrin-expressing LC, EPH/ephrin-loaded albumin spheres could, hence, possibly represent a novel therapeutic alternative.

Core–shell nanoparticles allow for effective and sustained drug delivery to tumors by offering the core protection from the surrounding environment, a versatile surface for targeting group binding, and an enhanced nanostructure bioavailability [48]. In LC treatment, core–shell nanoparticles have been successfully employed for simultaneous chemotherapy and radiation sensitization [49,50]. Interestingly, EPHs/ephrins might act as targeting moieties on these nanoparticles and improve their therapeutic efficacy against LC cells.

Taspine is an alkaloid that acts as a potent cicatrizant [51]. In addition to its other various biological properties, taspine, along with its derivatives, exhibits profound anti-cancer effects in diverse cancer entities, including LC [52,53,54,55]. After Dai et al. described that taspine derivative 12k suppresses colorectal cancer cell growth by competitive targeting of ephrin-B2-related pathways, EPH/ephrin-expressing LC also arose as a feasible target of taspine anti-tumor treatment [56].

The main available therapeutics targeting the EPH/ephrin system in LC are depicted in Table 1.

## 3. EPHA-Targeting Therapeutic Agents

A large number of original research articles has, to date, been published on EPHA-targeting treatment agents.

Bai et al. assessed multiple LC patient cohorts and defined *EPHA* mutation as an anti-programmed death-1 (PD-1) ligand 1 (PD-L1) efficacy predictor, given the prolonged progression-free survival (PFS) and overall survival (OS) of *EPHA*^mut^ lung adenocarcinoma patients. Specifically, *EPHA*^mut^ correlated with higher T-cell signatures and reduced transforming growth factor (TGF)-β signaling in those patients [57].

EPHA2 represents the most studied member of the EPH/ephrin family in LC therapy. So far, several study groups have reported on the association of EPHA2 with EGFR kinase inhibitors in LC. Li et al. reported strong EPHA2 enrichment by dasatinib [58], while Choi et al. not only confirmed that both gefitinib and erlotinib reciprocally regulated EPHA2 expression in TKI-sensitive LC cells, but also treated *EGFR*-mutant HCC827 cells with the recombinant EPHA2-Fc peptide, thus inhibiting anchorage-independent NSCLC cell growth [59]. Furthermore, Koch et al. employed kinase affinity purification and quantitative mass spectrometry and proposed high EPHA2 expression in gefitinib-resistant HCC827 LC cells. Gefitinib sensitivity was only restored after siRNA-mediated EPHA2 knock-down or dasatinib application, which lessened focal adhesion kinase (FAK) phosphorylation and cell migration [60]. Srivastava et al. showed that the EPHA2 TKI ensartinib may synergistically act with the roundabout guidance receptor 1 (ROBO1) ligand slit guidance ligand 2 (SLIT2) to attenuate the growth of squamous LC cells, which exercises its tumor-suppressive effect by either EPHA2 heterodimerizing or homodimer binding and inhibits AKT activity [61]. Interestingly, Amato et al. described EPHA2 overexpression in erlotinib-resistant LC cells, and suggested loss of *EPHA2* after genetic ablation in vitro and gene targeting in vivo, to account for apoptosis induction, decreased proliferation, as well as LC growth repression. Notably, the use of the EPHA2 small-molecule inhibitor ALW-II-41-27 showed similar results [62].

Ishigaki et al. presented cell-cycle arrest through the dephosphorylation of retinoblastoma (Rb) as the underlying mechanism of the proliferation inhibition of EPHA2-expressing SCLC cells, after both genetic siRNA EPHA2 inhibition and ALW-II-41-27/dasatinib treatment [63]. Additionally, Kaminskyy et al. targeted EPHA2 expression with specific siRNA in NSCLC cell lines, and highlighted the amplification of apoptotic signaling after an EPHA2 knockdown combination with ionizing radiation due to the partial phosphorylation reduction of the DNA-dependent protein kinase catalytic subunit (DNA-PKcs) [64].

Two independent research groups also investigated the plausibility of nanoparticle administration in EPHA2-expressing LC. Iyer et al. generated dual-stimuli nanoparticles (E-DSNPs) loaded with cisplatin and a radiation sensitizer and functionalized with anti-EPHA2 antibodies, which selectively targeted EPHA2-expressing NSCLC cells. E-DSNPs demonstrated a triggered release of the radiosensitizer during concurrent radiation therapy, followed by chemo-drug release upon glutathione exposure, thus downregulating the in vitro LC cell survival fraction [65]. Based on the assumption that the EPHA2–ephrin-A1 interaction attenuates NSCLC growth and survival, Lee et al. used albumin mesosphere-conjugated ephrin-A1 in vivo, and suggested apoptosis induction, along with tumor shrinkage, in mouse NSCLC xenograft models [66].

Gan et al. launched a phase I clinical trial of the anti-EPHA2 DS-8895a antibody in patients with advanced EPHA2-expressing epithelial cancers, one of whom was diagnosed with NSCLC. Although DS-8895a was well tolerated at the evaluated doses, the ^89^Zr trace-labelled infusion of DS-8895a demonstrated specific low-grade tumor uptake, while no objective tumor responses were noticed [67].

So far, two study groups have assessed the role of EPHA3 in LC treatment. Peng et al. explored the role of EPHA3 in multidrug resistance and observed EPHA3 overexpression to be associated with reduced phosphorylation of the PI3K/BMX/STAT3 signaling pathway, cell-cycle arrest, apoptosis induction, as well as chemosensitivity in SCLC [68]. Furthermore, Sos et al. grew *EPHA3*-amplified HCC515 NSCLC cells and demonstrated significant in vivo NSCLC shrinkage after dasatinib treatment [69].

Staquicini et al. created the anti-EPHA5 mAb 11C12 and suggested that 11C12 sensitized LC cells to radiotherapy both in vitro and in vivo, and showed a combined anti-proliferative/pro-senescence effect, which significantly prolonged OS [70].

By siRNA-mediated EPHA7 silencing, Li et al. managed to inhibit NSCLC cell proliferation, migration, and invasion; enhance B-cell lymphoma 2 (Bcl-2) -associated X protein, caspase-3, and phosphatase and tensin homolog (PTEN) expression levels; as well as downregulate phosphorylated-AKT levels [71].

Altogether, EPHA2, EPHA3, EPHA5, and EPHA7, seem to represent feasible targets for newly developed targeted treatment agents in (N)SCLC. Therapeutic up- or downregulation of EPHA2, EPHA3, EPHA5, and EPHA7, not only directly influence complex molecular pathways related to LC cell survival and progression, but also enhance the therapeutic efficacy of well-established treatment regimens (chemotherapy/radiotherapy/targeted therapy) for advanced LC.

## 4. EPHB-Targeting Therapeutic Agents

EPHB4 represents the most studied EPHB in LC treatment. Ferguson et al. first described that the combination of siRNA-mediated EPHB4 knockdown with topoisomerase I inhibition diminished SCLC cell viability in vitro, while, when co-applied with paclitaxel, inducing tumor regression in vivo [72]. In NSCLC, EPHB4 was found to be significantly downregulated by osimertinib [73].

Yoon et al. underlined that *EPHB6* mutation induces cell adhesion-mediated paclitaxel resistance via *EPHA2* and *CDH11* expression in LC and proposed ALW-II-41-27 therapy to suppress *EPHB6* (Q926R)-induced *CDH11* expression and focal adhesion formation [74].

All in all, therapeutic downregulation of EPHB4 and EPHB6 seems to specifically improve paclitaxel efficacy in LC.

## 5. Ephrin-Targeting Therapeutic Agents

Ephrin-A1, -A2, -B2, and -B3, have, to date, been suggested as feasible targets of treatment agents in LC.

In 2011, Lee et al. published their first research paper on targeted LC therapy using ephrin-A1-loaded albumin microspheres and underlined that albumin microspheres exhibited low toxicity for A549 cells, as well as rapid phagocytosis after incubation. Of note, these microspheres effectively reduced FAK expression, NSCLC cell migration, and tumor growth in matrigels [75]. Two years later, the same study group published their second article on the targeted delivery of highly stable, low-cytotoxicity let-7a microRNA-encapsulated ephrin-A1-conjugated liposomal nanoparticles in LC, which resulted in repressed *Ras* expression, as well as reduced NSCLC proliferation, migration, and tumor growth [76]. By achieving this breakthrough, Lee and coworkers reached a milestone and they still remain the first and only research group to have employed RNA-based therapeutics targeting the EPH/ephrin system in LC treatment. In this context, Murugan et al. recently published their comprehensive review article on up- and downregulated microRNAs in LC, and outlined the current advancements in the development of relevant microRNA-based therapeutics [77]. Importantly, the authors focused on the compelling need to ensure tumor tissue-specific delivery, given the unwanted off-target effects of microRNA-based suppression and replacement therapy, and accentuated the fact that, to date, except for the numerous microRNA delivery strategies in preclinical stages, there are only two registered clinical trials but no FDA-approved microRNA LC therapeutics [77].

Huang et al. generated the ephrin-A2-targeted taxane liposomal prodrug 2. This immunoliposome displayed an equilibrium dissociation constant toward the extracellular ephrin-A2 receptor domain, while its application in the NSCLC xenograft model A549 resulted in a profound tumor regression [78].

In addition, Dai et al. reported that the taspine derivative 12k repressed the ephrin-B2 and its PDZ protein, thus impairing vascular endothelial growth factor receptor (VEGFR)-2/-3 expression and suppressing A549 cell migration [79].

Finally, siRNA-mediated ephrin-B3 silencing led to decreased NSCLC U-1810 cell proliferation, as well as a morphological change from a small-round to a flattened-elongated appearance [80].

Taken altogether, ephrin-A1, and -A2, seem to represent ideal targets for drug delivery in LC, while pharmacologic downregulation of ephrin-B2, and -B3, directly suppresses LC cell proliferation and migration.

The effects of different therapeutic agents on the EPH/ephrin system in LC are summarized in Table 2.

## 6. Conclusions

Given its high prevalence and mortality, LC has long been the focus of diverse studies on the development of potent anti-cancer drugs. Despite the therapeutic advancements over the past 20 years, which have undoubtedly revolutionized anti-LC treatment, 5-year relative survival rates amount to 26% and 7% for NSCLC and SCLC, respectively, for all Surveillance, Epidemiology, and End Results (SEER) stages combined [81]. Consequently, the generation and establishment of effective novel therapeutic regimens are of utmost importance. In this context, the EPH/ephrin system seems to represent an achievable and most promising target for anti-LC treatment, and numerous research groups have already outlined the feasibility of EPH/ephrin targeting in other cancer entities [12,15,16,18]. In this review, we were able to identify EPHA2, EPHA3, EPHA5, EPHA7, EPHB4, EPHB6, ephrin-A1, ephrin-A2, ephrin-B2, and ephrin-B3 as potential targets of novel therapeutic agents, alone or in combination with well-established therapies (Figure 1 and Figure 2), and to demonstrate that their pharmaceutical up- or downregulation may lead to profound (N)SCLC suppression both in vitro and in vivo. Most significantly, EPH/ephrin targeting either restituted or improved LC cell sensitivity to chemo-, radio-, and targeted therapy, thus minimizing the quota of LC patients not profiting from already-FDA-approved treatments, which experts have sufficient experience with. Unfortunately, the only clinical trial incorporating a patient with advanced NSCLC could not describe objective anti-tumor effects after anti-EPHA2 DS-8895a antibody administration, hence questioning the results of the preclinical studies. As such, further clinical studies using larger patient collectives need to be conducted in order to verify the clinical utility and safety of EPH/ephrin-targeting agents in LC. Ideally, more emphasis should be given on the more aggressive SCLC subtype, which correlates with a worse prognosis and OS [81]. In summary, EPHs/ephrins seem to represent novel auspicious targets in modern LC therapy.

## Figures and Tables

**Figure 1 ijms-24-00093-f001:**
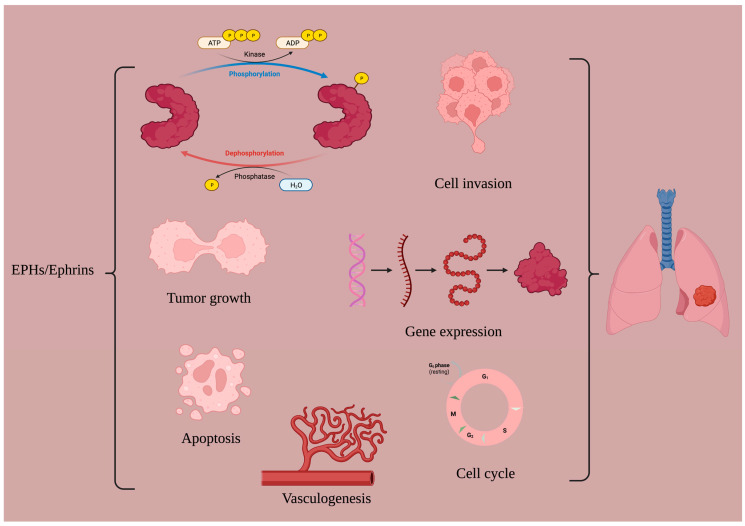
The role of the EPH/ephrin system in LC. Created with BioRender.com.

**Figure 2 ijms-24-00093-f002:**
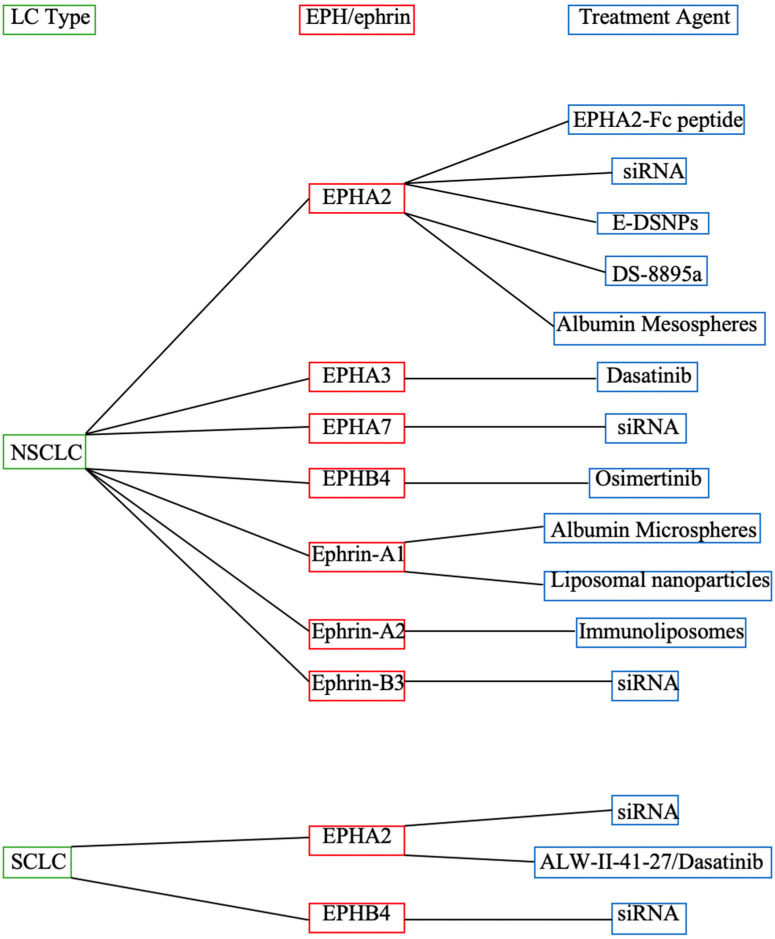
Treatment agents targeting the EPH/ephrin system in LC.

**Table 1 ijms-24-00093-t001:** Different therapeutics targeting the EPH/ephrin system in LC.

Targeting Strategy	Mechanism of Action
Small-molecule drugs	Tyrosine kinase inhibitors
Regulatory small RNA molecules	Short interfering RNAs
Monoclonal antibodies	Anti-EPH/ephrin monoclonal antibodies, antibody–drug conjugations
Immunoliposomes	Antibodies coupled to the liposomal surface
Nanoparticles	Albumin microspheres, core–shell nanoparticles
Natural compounds	Taspine derivatives

**Table 2 ijms-24-00093-t002:** Effects of diverse therapeutic agents on the EPH/ephrin system in LC.

Targeted EPH/Ephrin	Therapeutic Agent	Mechanism of Action	References
EPHA2	TKIs	EPHA2 Enrichment; Reciprocal EPHA2 expression regulation; FAK phosphorylation reduction; AKT activity downregulation; Cell migration inhibition; Apoptosis Induction; Decreased Proliferation; Tumor growth repression.	[58,60,61,62]
	EPHA2-Fc peptide	Anchorage- independent NSCLC cell growth inhibition.	[59]
	siRNA	Rb dephosphorylation; Cell-cycle Arrest; Partial DNA-PKcs phosphorylation; reduction; Apoptotic signaling amplification.	[63,64]
	E-DSNPs	Triggered release of the radiosensitizer during concurrent radiation therapy; Chemo-drug release upon glutathione Exposure; In vitro LC cell survival fraction downregulation.	[65]
	Albumin mesosphere-conjugated ephrin-A1	Apoptosis Induction; NSCLC in vivo tumor Shrinkage.	[66]
	DS-8895a	Low-grade tumor uptake.	[67]
EPHA3	Dasatinib	NSCLC in vivo tumor Shrinkage.	[69]
EPHA5	11C12	Sensitization to Radiotherapy; Anti-proliferative/pro- senescence effect; Prolonged OS.	[70]
EPHA7	siRNA	Inhibition of NSCLC cell proliferation, migration, and invasion; Enhancement of Bcl-2- associated X protein, caspase-3, and PTEN expression levels; Phosphorylated-AKT level downregulation.	[71]
EPHB4	siRNA	Reduced SCLC cell viability in vitro; Tumor regression in vivo.	[72]
	Osimertinib	EPHB4 downregulation.	[73]
EPHB6	ALW-II-41-27	*EPHB6* (Q926R)- induced *CDH11* expression downregulation; Suppressed focal adhesion formation.	[74]
Ephrin-A1	Ephrin-A1-loaded albumin microspheres	Reduced FAK expression; NSCLC cell migration Inhibition; Tumor growth repression.	[75]
	let-7a microRNA-encapsulated ephrin-A1-conjugated liposomal nanoparticles	Repressed *Ras* expression; Reduced NSCLC proliferation; Cell migration inhibition; Tumor growth arrest.	[76]
Ephrin-A2	Ephrin-A2-targeted taxane liposomal prodrug 2	In vivo tumor regression.	[78]
Ephrin-B2	12k	VEGFR-2/-3 downregulation; Cell migration inhibition.	[79]
Ephrin-B3	siRNA	Decreased cell proliferation; Morphological changes.	[80]

## Data Availability

Not applicable.

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
