# Peer review of "May EPH/Ephrin Targeting Revolutionize Lung Cancer Treatment?"

_ijms, 2022, doi:10.3390/ijms24010093_

Round 1
Reviewer 1 Report
The authors present the EPH/Ephrin system as a target for lung cancer therapy, but the review in this form is not very convincing and clear, being very general in its data presentation. A graphical representation of the various molecular interactions should also be inserted to make it more readable and understandable by readers.
In this form the article is not publishable and should be improved.
Author Response
We would like to thank the reviewer for his constructive proposal. We have now added Figure 1 that graphically presents the various molecular interactions in order to make the review more readable and understandable by readers. Notably, the intention of this review is to summarize and concisely present all currently available therapeutic agents targeting the EPH/ephrin system in LC. By presenting all important key factors of each agent, but meanwhile avoiding unnecessary details that would confuse the readership, we aimed at composing a structured review that to our knowledge represents the most comprehensive uptodate review of the literature on the role of EPH/ephrin targeting in LC therapy.
Reviewer 2 Report
This kind of work is very much needed in molecular biology research, so I reviewed this article on a priority basis. The author’s approach in the manuscript is well written, with a scientific temperament and relevant references. The authors have consolidated the study by drawing an appropriate conclusion based on the collected data, but the manuscript misses the role of the non-coding RNAs (miRNA, lncRNA, circRNA, and piRNA) targeting EPHA.
Comments:
RNA therapeutics moved from unrealistic dreams to genuine realities. To date, around 11 microRNA-based drugs have been approved by the FDA. Approximately 20 clinical trials have been initiated using miRNA-based therapeutics in phases II and III of clinical development. The authors should add a paragraph describing the new advances in RNA-based therapeutics targeting the EPH/ephrin system in lung cancer treatment.
Author Response
We would like to thank the reviewer for this constructive comment. We have now added a paragraph describing the new advances in RNA-based therapeutics targeting the EPH/ephrin system in lung cancer treatment.
Round 2
Reviewer 1 Report
If it possible, the quality of figure 1 should be improved.